# PaleoRiada: A new integrated spatial database of palaeofloods in Spain

Kelly Patricia Sandoval-Rincón<sup>1</sup>, Julio Garrote-Revilla<sup>2</sup>, Daniel Vázquez-Tarrío<sup>1</sup>, Silvia Cervel<sup>1</sup>, Jose Hernández-Manchado<sup>1</sup>, Juan López-Vinielles<sup>1</sup>, Rosa María Mateos<sup>1</sup>, Juan Antonio Ballesteros-Cánovas<sup>3</sup>, Gerardo Benito<sup>3</sup>, Andrés Díez-Herrero<sup>1</sup>

Correspondence to: Andrés Díez-Herrero (andres.diez@csic.es)

Abstract. Palaeoflood records are natural evidence of past flood events (typically found in landforms, sediments, or vegetation). Over the last 25 years, several palaeoflood record databases have been implemented. However, many of these databases are outdated, lack accessible or comprehensive palaeohydrological information, and present challenges in terms of data accessibility and reuse, particularly for non-research communities (e.g. planning administrations or flood risk managers). This work introduces PaleoRiada, the first open database that compiles published palaeoflood records from Spain. PaleoRiada stores typological, hydrological, temporal, and spatial data collected from approximately 126 publications (including journal articles, scientific reports, and book chapters). This database has been implemented using a simple Relational Database Management System (RDMS), integrated into a web platform, and is freely accessible at https://zenodo.org/records/17391823 (Sandoval-Rincón et al., 2025). The PaleoRiada database contains 299 palaeoflood records (both geological and biological) dated between 2014 CE and 97,000 BP and distributed across both Atlantic (164) and Mediterranean (135) catchments. PaleoRiada includes 157 records with specific discharge values ranging from 0.02 to 320 m<sup>3</sup>·km<sup>-2</sup>·s<sup>-1</sup>. These records are associated with a variety of river systems, including wide alluvial plains (25), Mediterranean ephemeral streams (17), mountain torrents (36), and confined valley rivers (79). Additionally, they encompass overbank flood events (102), flash floods (48), dam failures (1), and hyperconcentrated flow events (6). The relationship between PaleoRiada and the Spanish Flood-prone Mapping Project (SNCZI) indicates that approximately 80% of the PaleoRiada records are not included in the flood-prone areas defined by SNCZI. Therefore, several records can be consulted to prioritise or propose new areas for preliminary flood risk assessment. Accessibility and simplified data query and entry in PaleoRiada can facilitate the application of palaeoflood data in land planning and flood risk management.

<sup>&</sup>lt;sup>1</sup>Department of Geo-Hazards & Climate Change, Geological and Mining Institute of Spain (IGME), Spanish Scientific Research Council (CSIC), Rios Rosas 23, Madrid, Spain

<sup>&</sup>lt;sup>2</sup>Department of Geodynamics, Stratigraphy and Palaeontology, Complutense University of Madrid, José Antonio Novais 12, Madrid, Spain.

O <sup>3</sup>Department of Geology, National Museum of Natural Sciences (MNCN), Spanish Scientific Research Council (CSIC), José Gutiérrez Abascal 2, Madrid, Spain.

## 1 Introduction

- Flood events are often recorded in instrumented series (gauging stations) and/or different kinds of documentation (texts, images, video recordings, plaques on buildings). Systematic data barely record recent floods (last century) (Malaak et al., 2011; Francés and Botero, 2022; Delus et al., 2023; Claps et al., 2024). Historical floods (last 2,000 years) are commonly recorded in documentary databases (Kjeldsen et al., 2014; Machado et al., 2015; Benito et al., 2015a; Boisson et al., 2022; Renard, 2023). However, many flood events can be unnoticed because: (i) they predate instrumented records and have been lost over time; (ii) they occurred in secondary (ungauged) rivers or exceeded the recording thresholds of gauging stations; (iii) they occurred in remote areas; or (iv) they did not cause significant socio-economic damage or losses (Díez-Herrero et al., 2023). Natural records offer alternative or complementary information to systematic and documentary flood data through palaeoflood analysis (Benito and Thorndycraft, 2005; Reinders and Muñoz, 2021; Benito et al., 2022).
- Palaeofloods refer to floods recorded in natural archives, specifically found in landforms, sediments, or vegetation left behind after significant flood events (Kochel and Baker, 1982; Baker, 1983; Baker, 2008). The term "palaeoflood" does not solely refer to a flood event from prehistoric times or before the Holocene, but rather to any past flood event evidenced in natural archives (Benito and O'Connor, 2013; Benito and Díez-Herrero, 2015). The palaeoflood concept was introduced in the 1960s (Sigafoos, 1961, 1964; Ballesteros-Cánovas et al., 2015a), and disseminated in the 1980s (Patton and Dibble, 1982; Ely and Baker, 1985; Chatters and Hoover, 1986; O'Connor et al., 1986; Webb et al., 1988). It is widely applied in contemporary research (Miller et al., 2019; Benito and Thorndycraft, 2020; Van Der Meulen et al., 2022; Guo et al., 2023).
  - Palaeoflood analysis generally involves estimating the ages and magnitudes of major floods (Baker, 2008; Benito and Díez-Herrero, 2015). This approach is key for improving flood frequency analysis (Harrison and Reid, 1967; O'Connell et al., 2002; Jenny et al., 2014), as it extends the temporal record and enables a more accurate estimation of flood quantiles at low annual exceedance probabilities (Benito et al., 2022). In this regard, the findings obtained from palaeoflood data can be used for: (i) studying the climate-flood relationship (Ely, 1997; Macklin and Lewin, 2003; Wilhelm et al., 2013; Benito et al., 2015b; Benito et al., 2023a); (ii) conducting engineering, design, and safety studies of critical structures (Levish et al., 1997; Benito et al., 2006; Cloete et al., 2022); and (iii) assessing flood hazard and risk (Levish, 2002; Benito et al., 2004; Benito et al., 2005; Liu et al., 2019).
- Despite the varied applications noted, comprehensive palaeoflood databases are still emerging, and most existing ones have limited access and updates (Hirschboeck et al., 1996; Branson, 1995; Casas Planes et al., 2003; Díez-Herrero et al., 1998). Over the last decade, international palaeoflood databases with accessible temporal data, such as the database of flood records from lake sediments in the European Alps (Wilhelm et al., 2022), and global collaborative flood databases, such as the PAGES-Floods Working Group Database (Wilhelm et al., 2019), have been developed. Nevertheless, these data banks are limited to a single typology of palaeoflood records or have low regional scope. At national and regional levels, palaeoflood data storage and publication have been little addressed over the last 20 years, and those published data typically consist of an event-age relational table without related spatial data. Examples of this include palaeoflood data from India (Kale, 2008) and the Colorado

River Basin (Enzel et al., 1993). In Spain, SPHERE-GIS (Casas Planes et al., 2003) and PaleoTagus (Díez-Herrero et al., 1998) are the only flood record repositories with palaeoflood data published to date. However, they have inaccessible data and metadata and restricted spatial coverage (Internal Basins of Catalonia and the Tagus River Basin).

All the aforementioned palaeoflood datasets lack palaeohydrological information (such as discharge, water levels, associated precipitation thresholds, flow and stream types) or do not make this information accessible. Furthermore, they have a user-unfriendly structure that makes the data difficult to find and reuse, particularly for non-research communities (e.g. planning administrations or flood risk managers). Considering these limitations and to address the gap in the availability of national palaeoflood databases in Spain, we have designed and implemented PaleoRiada: The Palaeoflood Records Database of Spain.

PaleoRiada aims to provide: (i) an open and georeferenced national dataset that is reusable by researchers, planning authorities, consulting firms, and the public; (ii) clear data and metadata to facilitate the retrieval of palaeohydrological information; and (iii) a tool with potential applications in national flood risk management (e.g. to prioritise or propose new areas for preliminary flood risk assessment).

PaleoRiada is the first publicly accessible database that compiles published palaeoflood records from various sources, including scientific journals, book chapters, and conference proceedings, from all over Spain. Currently, PaleoRiada stores 299 palaeoflood records (biological and geological), of which 197 include hydrological information. PaleoRiada has been implemented using a Relational Database Management System (RDMS) and is hosted on an intuitive web application to foster broader usability.

In this article, we provide the dataset and outline the structure of the PaleoRiada database, detailing its data sources and collection methodology. We also present a thematic, spatial, and temporal description of the dataset, while discussing its applications, advantages, and limitations.

#### 2 Database structure

The structure of the PaleoRiada database was designed based on a relational data model. The methodology applied to implement PaleoRiada included the prior and sequential design of the conceptual, logical, and physical models (Codd, 1970; Watt and Eng, 2014). The conceptual model establishes the foundation of the structure through the definition of entities and information requirements, while the logical model defines groups of tables and the relationships between them. The physical model is created by transferring the logical model to a database management system, enabling the creation of the necessary tables, fields, relationships, and keys to ensure the integrity of the information.

# 2.1 Conceptual Model

PaleoRiada has been designed from a conceptual model encompassing three groups of entities describing the sources of information and thematic and spatial characteristics of palaeoflood records. These three groups are: (i) Basic Data-related Entities (BDE); (ii) Geospatial-related Entities (GSE); and (iii) Hydrology Information-related Entities (HIE) (Fig. 1). Through the first two groups of entities, we aim to store temporal, typological, and spatial information in each database entry (as a minimum requirement). Additionally, the third group of entities (optional) allows for incorporating hydrological information into each record. Each database entry includes general geographic information (associated river stream, region, hydrographic basin, etc.) and a relationship specifying the record's location via a point-type vector file.

Figure 1: Conceptual Schema of the PaleoRiada Database. BDE: Basic Data-related Entities; GSE: Geospatial-related Entities; HIE: Hydrology Information-related Entities

#### 2.2 Logical Model

Once the conceptual model of the database structure was established, it was translated into a logical model (Date, 1999; Codd, 1972), where entities become tables, attributes become fields, and relationships between tables are assigned through primary and foreign keys (Napolitano et al., 2018). In the case of PaleoRiada, the entity groups BDE, GSE, and HIE are associated with the following groups of tables, respectively: (i) BIT: Basic Information Tables; (ii) SIT: Spatial Information Tables; (iii) HIT: Hydrology-related Tables. A detailed description of these groups of tables, their main fields, and the relationships that make up the proposed logical model is provided below.

# 115 **2.2.1 The BIT Group**

140

The BIT tables store alphanumeric information describing the palaeoflood record in the following terms: (i) geographic (administrative and hydrographic units where the record and its sampling points are located); (ii) typological (type and description of the evidence used to identify and date the record); (iii) temporal (age of the palaeoflood evidence related to the record, dating method, uncertainty in age estimation, among others); and (iv) bibliographic (sources, authors, and publication year of the records). Within this group of tables, the identifier field of the palaeoflood records (ID) contains the primary key (PK) for each record entered into the database. This primary key links the palaeoflood records to the information stored in the SIT and HIT groups of tables.

# 2.2.2 The SIT Group

The SIT tables contain relational and spatial information (point coordinates) that locate the records. These tables link the sampling points (stored in the vector file) with the palaeoflood records. The relationship between the SIT and BIT table groups was assigned considering that the points stored in the spatial file could be associated with multiple records, as several successive palaeoflood records can exist at the same sampling point.

## 2.2.3 The HIT Group

The HIT tables store data on discharges, estimated levels and velocities, and specific information extracted from publications (precipitation data, estimation methods, and uncertainty of discharge values). The HIT tables are related to the palaeoflood records through an N:1 relationship. This HIT-BIT relationship was assigned considering that a palaeoflood record can have hydrological-hydraulic data from more than one information source or estimation method.

## 2.3 Physical Model

After defining the groups of tables, fields, and their relationships (established in the logical model and detailed in Appendix A), we implemented the physical model of the database structure (Watt and Eng, 2014). The physical model of PaleoRiada was implemented using the Microsoft Access database management system, creating: (i) data entry tables; (ii) association tables; (iii) dictionary tables; (iv) fields for each table; and (v) primary (PK) and foreign keys (FK).

The physical model of PaleoRiada comprises 16 tables (**Fig. 2**), including 3 data entry tables that store general record data (Palaeoflood Record table), spatial data (Point table), and hydrological data (Hydrological Information table). The remaining 12 tables are dictionaries created to fill in the table fields and protect referential integrity through primary keys (Napolitano et al., 2018), as well as one intermediate table used to include relationships between palaeoflood records and location points.

Figure 2. PaleoRiada Database Physical Structure. BIT: Basic Information Tables, SIT: Spatial Information Tables, HIT: Hydrology-related Tables

Finally, to facilitate the visualisation and querying of PaleoRiada data, we developed a web application in ArcGIS Online using the Web AppBuilder tool for customised interactive mapping, querying, and exploration of the dataset's spatial patterns (https://sgaicsic.maps.arcgis.com/apps/webappviewer/index.html?id=047e3ddb46354f5785e59ff06c2cd5eb). This web application includes a data entry form. In this regard, information on new records or palaeoflood studies conducted in Spain can be incorporated into a temporary layer by any user and subsequently moved to a review phase by the database managers. The PaleoRiada database (tables and metadata) is freely available for download on Zenodo via the following DOI: https://zenodo.org/records/17391823 (Sandoval-Rincón et al., 2025).

## 2.4 Data sources and information collection

150

- The collection of alphanumeric information stored in the current PaleoRiada dataset was conducted by consulting approximately 126 texts published between 1996 and 2024. The bibliographic sources included journal articles (117), scientific reports (4), and book chapters (5). All these sources were selected through a systematic search for palaeoflood publications in each of Spain's main hydrographic basins using databases and search engines, such as Scopus, ScienceDirect, Web of Science, SpringerLink, and Google Scholar.
- To obtain the temporal, typological, hydrological, and spatial information of the records entered into the PaleoRiada database, a detailed review was conducted of both the results cited in the text and the data presented in tables (e.g. dating values) and figures (e.g. stratigraphic columns, hydraulic models, channel sections, location maps, etc.). This review was supported by the bibliographic tool Zotero (www.zotero.org). Using Zotero, the alphanumeric data of each record were collected through citations, notes, and queries. Each palaeoflood record was simultaneously consulted across all publications, thus considering any possible updates to the published information. All data sources consulted and extracted information from them are published and accessible through scientific databases or institutional repositories. Consequently, the data stored in PaleoRiada presents no ownership or authorship conflicts, provided that the sources are properly referenced. To this end, the database includes a field for the reference of the first published work related to each record, along with another field for subsequently developed works.

## 170 3 PaleoRiada dataset description

PaleoRiada currently includes a total of 299 georeferenced palaeoflood records, 197 of which contain hydrological information. These data are spatially linked to a vector file and can be easily viewed and queried through a web application or GIS. Below is a more detailed description of the current PaleoRiada dataset.

# 3.1 Temporal data

From a temporal perspective, the PaleoRiada dataset spans 97,064 years (97,000 BP to 2014 CE). Eighty-five percent of the total records date within the Common Era (CE), with at least one record in each century (except for the 2nd and 4th centuries). Furthermore, the highest number of records dated within the CE corresponds to the years 1855, 1955, 1973, 1997, and 2005

(**Fig. 3**). On the other hand, the remaining 15% corresponds to palaeoflood records from before the Common Era (BCE), distributed within the time range of 2060 to 97,000 years BP (**Table 1**).

The temporal difference between chronologically consecutive records shows that BCE palaeoflood records are, on average, 2260 years apart, with the 75th percentile being 545 years. In contrast, CE records are approximately 8 years apart on average, with 55% of the differences being one year or zero (**Table 1**). This indicates that using the PaleoRiada dataset for long-term natural climate variability analysis may require supplementation with additional information sources (Benito et al., 2020; Archer et al., 2017).

Figure 3. Temporal distribution of the PaleoRiada records dated within the Common Era

Table 1. Frequency of Palaeoflood records Before the Common Era (left). Statistics describing the temporal continuity of BCE and CE record sets (right).

|             | BCE record | ds frequency  |       | Temporal continuity             |                |               |  |
|-------------|------------|---------------|-------|---------------------------------|----------------|---------------|--|
| BP years    | Freq.      | BP years      | Freq. | Temporal difference* statistics | BCE<br>records | CE<br>records |  |
| 2000 - 3000 | 6          | 9000 - 10000  | 5     | Min                             | 0.0            | 0.0           |  |
| 3000 - 4000 | 5          | 10000 - 11700 | 4     | Max                             | 51764          | 200           |  |
| 4000 - 5000 | 5          | 11700 - 15000 | 5     | Mean                            | 2260.48        | 7.81          |  |
| 5000 - 6000 | 1          | 15000 - 45000 | 2     | Standard deviation              | 8464.39        | 25.08         |  |
| 6000 - 7000 | 1          | 45000 - 60000 | 1     |                                 |                |               |  |
| 7000 - 8000 | 3          | >60000        | 2     | 50th percentile                 | 249            | 1.0           |  |
| 8000 - 9000 | 4          |               |       | 75th percentile                 | 545            | 3.0           |  |
|             |            | Total records | 44    | * Be                            | etween consec  | utive recor   |  |

The ages of palaeoflood records in the PaleoRiada dataset were estimated using a variety of dating approaches (**Table 2**). The most frequent methods are radiometric (38% of records) and dendrochronological (35%) techniques, followed by luminescence dating (11%) and documentary sources (10%), the latter in some cases calibrated or cross-checked with absolute dating methods. Less common are archaeological dating (4%), microscopy (0.3%), and lichenometry (0.3%). This diversity of approaches reflects both the range of environments where palaeoflood evidence is preserved and the variety of temporal resolutions achievable across the dataset.

The relative uncertainty of the ages assigned to PaleoRiada records is strongly influenced by the record typology and the correlation analyses applied in the original studies. For example, some geological records have specific dates derived from correlations with documentary information (ages between 1617 and 2013 CE), while others have ages estimated from archaeological dating (ages between 2088 BP and 1856 CE). The dataset also exhibits a temporal bias, primarily related to the preservation range of biological records (last three centuries) and the predominance of certain dating methods, for instance, the marked increase in records dated to 1955 due to the widespread use of <sup>137</sup>Cs dating (**Fig. 3**).

Table 2. Distribution of dating methods in the PaleoRiada dataset

| Dating Method  | Record Freq. | % dataset | Dating Method       | Record Freq. | % dataset |
|----------------|--------------|-----------|---------------------|--------------|-----------|
| Documentary*   | 30           | 10%       | Luminescence        | 34           | 11%       |
| Archaeological | 12           | 4%        | Dendrochronological | 106          | 35%       |
| Radiometric    | 115          | 38%       | Lichenometry        | 1            | 0.3%      |
| Microscopy     | 1            | 0.3%      |                     |              |           |

<sup>\*</sup>Specific dates based on correlations with documentary information.

The PaleoRiada dataset contains explicit numerical age-uncertainty values (in years) only for records dated using radiometric methods, luminescence techniques and lichenometry (**Table 3**). These uncertainties range from as little as ±4 years for very recent lichenometric or <sup>210</sup>Pb-based dates (e.g. late 20th–21st century CE) to as much as ±6,000 years for the oldest OSL-dated records (44,736–97,000 BP). Radiocarbon (<sup>14</sup>C) dating, which in the database has been applied to both BCE and CE records, shows a comparable pattern: uncertainties are smallest (±30–95 years) for more recent samples (30–1,947 CE) and largest (±1,860 years) for older samples (2,640–32,600 BP). This indicates that, within the dataset, age uncertainty is governed primarily by the antiquity of the sample rather than by the dating method alone. Users requiring high temporal precision can make use of the uncertainty field (IDT in the Record Basic Data Table) to filter and select the most appropriate records for frequency or occurrence analyses. Notably, no records were excluded from PaleoRiada due to high age uncertainty, to preserve the completeness of the published evidence; instead, uncertainty information is provided so that each user may apply selection criteria suited to their specific needs.

Table 3. Dating submethods, and associated temporal range and age uncertainty of records in the PaleoRiada dataset

| Dating Submethod              | Record age range   | Minimum<br>uncertainty* | Maximum uncertainty | Mean<br>uncertainty |
|-------------------------------|--------------------|-------------------------|---------------------|---------------------|
| OSL Luminescence              | 44,736 – 97,000 BP | 2900                    | 6000                | 4066                |
| <sup>14</sup> C Radiometric   | 2,640 – 32,600 BP  | 30                      | 1860                | 237                 |
| U/Th Radiometric              | 1709 – 1755 CE     | 66                      | 70                  | 68                  |
| <sup>14</sup> C Radiometric   | 30 – 1947 CE       | 30                      | 95                  | 51.61               |
| <sup>210</sup> Pb Radiometric | 1902 CE            | 4                       | 4                   | 4                   |
| Thermoluminescence            | 1572 CE            | 55                      | 55                  | 55                  |
| Lichenometry                  | 1997 CE            | 4                       | 4                   | 4                   |

<sup>\*</sup>Uncertainty values are expressed in  $\pm$  years.

## 3.2 Spatial and typological characteristics

All palaeoflood records are linked to 80 sampling sites distributed between Spain's two main hydrographic catchments (Atlantic and Mediterranean) (Fig. 4). Regarding the distribution of records across rivers, the PaleoRiada dataset includes records located in 45 different water courses. There are palaeoflood records associated with wide alluvial plain rivers (25), Mediterranean ephemeral streams (36), mountain torrents (102), and confined valley rivers (109). Additionally, the PaleoRiada dataset includes 20 records from lakes (Arreo, Montcortés, Sanabria, and others) and 7 records from caves (Rosa, Ojo de Valjunquera, El Soplao, and others). Regarding typology, palaeoflood evidence in PaleoRiada consists of slack-water deposits (52%), tree scars and growth anomalies (36%), and a smaller proportion of erosional scars (e.g. sigmoid profiles and bank undercuttings; Bodoque et al., 2011; Jarrett and Tomlinson, 2000) and lichens (Table 4). The PaleoRiada dataset has a typological bias, primarily related to the preservation capacity of the evidence (e.g. the predominance of geological records in the 50-97,000 years BP interval; Fig. 5).

Table 4. Percentage distribution of records by palaeoflood evidence typology.

| Type       | Evidence Sub-type             | % data-set          | Evidence Sub-subtype             | % data-set |  |
|------------|-------------------------------|---------------------|----------------------------------|------------|--|
| al         | D 1 1111                      |                     | Wood accumulation                | 0.3%       |  |
| Biological | Dendrogeomorphological        | 36.5%               | Scars and growth anomalies       | 36.2%      |  |
| Biol       | Lichenometric                 | 0.3% Size of thalli |                                  | 0.3%       |  |
| Geological | Fluvial Sedimentological      | 54.2%               | Slack-water flood deposits (SWD) | 52.2%      |  |
|            |                               |                     | Boulder and cobble deposits      | 0.7%       |  |
|            | Lacustrine Sedimentological   | 6.7%                | Varved deposits                  | 6.4%       |  |
|            | Fluvial Geomorphological 0.2% |                     | Erosional scar                   | 0.2%       |  |
|            | Speleothem                    | 2.3%                | Speleothem                       | 2.3%       |  |
|            |                               |                     |                                  |            |  |

Figure 4. Map of record sampling sites. Locations with the highest record density are highlighted with white rectangles, with the number of records shown in brackets.

Figure 5. Record frequency by typology and age [BCE in years BP; CE in centuries] differentiated by basin.

Fluvial sedimentological evidence in PaleoRiada shows the widest variety of dating approaches (luminescence, radiocarbon, archaeological, and documentary correlations) and the highest average age uncertainty (± 293.8 years), mainly because they include the oldest events in the dataset. Some of these records were dated using OSL luminescence, with uncertainties exceeding 3,000 years, reflecting the inherent difficulty in precisely constraining the age of long-preserved deposits. This pattern is not attributable to a specific hydrographic basin: uncertainty values do not display a consistent trend. Instead, spatial differences arise from where research projects have historically been conducted, often determined by site accessibility, geomorphological suitability for evidence preservation, and the availability of datable material. As a result, similar types of evidence, and thus similar uncertainty levels, occur across different basins, indicating that uncertainty is more effectively explained by typology and the antiquity of the preserved evidence than by basin location.

#### 3.3 Hydrological data

The PaleoRiada dataset currently holds 197 records with hydrological information, including: i) 117 records with minimum discharge values, the majority of which (103) were estimated from fluvial sedimentological evidence using one-dimensional hydraulic models; ii) 30 records with maximum discharge values, primarily derived through empirical hydrological methods; and iii) 10 records with discharge values measured directly at gauging stations near the evidence. The remaining 40 records do not have estimated discharge values but include other hydrological information, such as precipitation thresholds, recorded discharges in nearby streams, and qualitative information on the magnitude of the event.

The range of maximum specific discharge values ( $Qs_{max}$ ) is broad (0.02 - 320 m<sup>3</sup>·km<sup>-2</sup>·s<sup>-1</sup>), reflecting the variety of flood types included in the database. Among the records with discharge data (157), there are: 102 records corresponding to floods occurring in confined rivers and rivers with wide alluvial plains, and 55 records related to flash floods occurring in mountain torrents, Mediterranean ephemeral streams, and confined rivers. Additionally, there is a single record associated with a flash flood caused by dam failure, and six records related to hyperconcentrated flows (**Table 5**). Finally, regarding the spatiotemporal distribution of discharge, a significant increase in  $Qs_{max}$  values was observed in the eastern and southern Mediterranean Basin during the 18th and 19th centuries (5.0 and 6.3 m<sup>3</sup>·km<sup>-2</sup>·s<sup>-1</sup>). These values stand out compared to the  $Qs_{max}$  values in the Atlantic and northeastern Mediterranean Basins (0.4 and 0.6 m<sup>3</sup>·km<sup>-2</sup>·s<sup>-1</sup>) during the same centuries (**Table 6**).

It is worth noting that the PaleoRiada dataset has a flood event magnitude bias. Over- or underestimation of discharge values may occur for several reasons. Many discharge estimates correspond to minimum water levels that could have been reached (e.g. slackwater deposits and tree scar records). Palaeoflood evidence is preferentially preserved for the largest extreme floods (Thorndycraft and Benito, 2006), meaning that smaller and more frequent events are typically absent. For long records, channel bed or base levels may have changed; however, most records in PaleoRiada are derived from relatively stable river sections, minimising this effect. These limitations should therefore be considered when using the PaleoRiada dataset for hydrological analyses. Similarly, when using the PaleoRiada discharge data for long-term analyses of floods and climate variability, it is essential to consider the differences between past and present land-use, which may cause substantial variations in flood magnitudes within the same basin (Feinberg et al., 2020; Schillereff et al., 2019).

Table 5. Maximum Specific Discharge Values (Qsmax) per Rivers/Streams (grouped by Flood Typology).

| Flood typology<br>River/Stream –Location | A<br>(km²) | Qs <sub>max</sub><br>(m <sup>3</sup> ·km <sup>-2</sup> ·s <sup>-1</sup> ) | Flood typology<br>River/Stream –Location | A<br>(km²) | Qs <sub>max</sub><br>(m <sup>3</sup> ·km <sup>-2</sup> ·s <sup>-1</sup> ) |
|------------------------------------------|------------|---------------------------------------------------------------------------|------------------------------------------|------------|---------------------------------------------------------------------------|
| Flash flood                              |            |                                                                           | Flood                                    |            |                                                                           |
| Arroyo Cabrera-Venero Claro              | 15.09      | 5.23                                                                      | Río Alberche-Navaluenga                  | 699.51     | 2.41                                                                      |
| Arroyo de las Pintadas-Valsaín           | 1.17       | 34.45                                                                     | Río Caramel-Rambla Mayor                 | 208.25     | 4.97                                                                      |
| Arroyo del P. del Paular- Valsaín        | 0.85       | 22.16                                                                     | Río Cega-Pajares                         | 237.63     | 0.29                                                                      |
| Barranco de Montardit-Montardit          | 7.00       | 16.00                                                                     | Río Duero-Bemposta                       | 63832      | 0.13                                                                      |
| Rambla de la Viuda-Rambla Viuda          | 1176       | 1.56                                                                      | Río Duratón-Duratón                      | 782.34     | 4.56                                                                      |
| Rambla Mayor-Rambla Mayor                | 144.88     | 11.15                                                                     | Río Eresma-Abrigo Molino                 | 244.98     | 6.02                                                                      |
| Río Arenal-Arenal Arenas                 | 50.70      | 7.14                                                                      | Río Guadalhorce-Gaitanejos               | 1681       | 1.71                                                                      |
| Río Pelayo-Pelayo                        | 6.31       | 16.00                                                                     | Río Guadalquivir-Marmolejo               | 19677      | 0.15                                                                      |
| Río Segre-Segre Alós                     | 4354       | 0.53                                                                      | Río Llobregat-Llobregat M.               | 3442       | 1.36                                                                      |
| Flash flood/Dam break                    |            |                                                                           | Río Llobregat-Llobregat V.               | 1889       | 2.28                                                                      |
| Río Tera-Vega Tera                       | 40.93      | 320.09                                                                    | Río Montlleó-Montlleó                    | 580.63     | 1.41                                                                      |
| Flash flood/Hiperconcentrated flow       |            |                                                                           | Río Segura-Segura A.Pozo                 | 6206       | 0.11                                                                      |
| Arroyo Cabrera-Venero Claro              | 4.20       | 141.09                                                                    | Río Tajo-Pta. Vado                       | 24525      | 0.13                                                                      |
| Barranco de Arás-Biescas                 | 19.42      | 22.14                                                                     | Río Tajo-Puente del Arzobispo            | 35135      | 0.09                                                                      |
| Barranco de las Angustias-Taburiente     | 5.07       | 100.50                                                                    | Río Tajo-Tajo Alcántara                  | 51772      | 0.23                                                                      |
| Barranco de Portainé-Portainé            | 5.71       | 55.38                                                                     | Río Ter-Ter                              | 2605       | 1.06                                                                      |

<sup>\*</sup>A refers to the upstream basin area of the sampling site.

Table 6. Maximum Specific Discharge Values (Qsmax) recorded in the PaleoRiada dataset grouped by time interval and basins.

| Date<br>type       | Atlantic           | Basin                                         | Mediterranean I    | Basin (E - S)                                       | Mediterranean Basin (NE) |                                               |  |
|--------------------|--------------------|-----------------------------------------------|--------------------|-----------------------------------------------------|--------------------------|-----------------------------------------------|--|
|                    | Date               | $Qs_{max}$ $(m^3 \cdot km^{-2} \cdot s^{-1})$ | Date               | $\frac{Qs_{max}}{(m^3 \cdot km^{-2} \cdot s^{-1})}$ | Date                     | $Qs_{max}$ $(m^3 \cdot km^{-2} \cdot s^{-1})$ |  |
|                    | 8 th               | 0.23                                          | 6 th               | 0.15                                                | 17 <sup>th</sup>         | 1.36                                          |  |
|                    | $10^{\mathrm{th}}$ | 0.08                                          | $10^{\mathrm{th}}$ | 0.11                                                | 18 <sup>th</sup>         | 0.44                                          |  |
| S                  | 11 <sup>th</sup>   | 0.08                                          | 14 <sup>th</sup>   | 1.69                                                | 19 <sup>th</sup>         | 0.58                                          |  |
| CE centuries dates | 12 <sup>th</sup>   | 0.04                                          | 15 th              | 1.31                                                | $20^{\mathrm{th}}$       | 22.14                                         |  |
|                    | 15 <sup>th</sup>   | 0.12                                          | $16^{th}$          | 3.27                                                | 21 st                    | 55.38                                         |  |
| entu               | 16 <sup>th</sup>   | 0.08                                          | $17^{\mathrm{th}}$ | 1.02                                                |                          |                                               |  |
| Œ                  | $17^{\text{ th}}$  | 0.05                                          | $18^{ th}$         | 5.04                                                |                          |                                               |  |
| O                  | 19 <sup>th</sup>   | 0.13                                          | 19 th              | 6.32                                                |                          |                                               |  |
|                    | $20^{\text{ th}}$  | 320.09                                        | $20^{ th}$         | 11.15                                               |                          |                                               |  |
|                    | 21 st              | 16.00                                         | 21 st              | 0.90                                                |                          |                                               |  |
| BCE                | 2.06-11.7 ky BP    | 0.13                                          | 2.06-11.7 ky BP    | 1.17                                                | 2.06-11.7 ky BP          | 2.28                                          |  |
| — <del>"</del>     | >11.7 ky BP        | 4.56                                          |                    |                                                     |                          |                                               |  |

Representative comparisons in terms of age and magnitude of flood events, and consistent analyses of climate variability-flood response, will require more in-depth data analysis, as PaleoRiada compiles hydrological data of records rather than single events.

Information on non-exceedance bounds and perception thresholds can be derived from the database even though the latter is not coded as a separate field. Non-exceedance bounds are only occasionally reported in the reviewed studies, usually inferred from the absence of depositional units or the preservation of stable soils on high terraces or surfaces. Although not systematically available, such evidence highlights intervals when flood stages did not exceed certain geomorphic thresholds, and it is documented in the Hydrological Information Table via the NEB field.

Perception thresholds, by contrast, can be reconstructed for most sites. These thresholds represent the minimum discharge or stage required to leave identifiable flood evidence at a depositional setting, and in some cases, they increase through time as sediment progressively builds up on flood benches (a self-rising component). In practice, perception thresholds can be obtained by combining the "minimum discharge" field with the contextual information provided in "other hydrological interpretation data" and the stratigraphic descriptions. Together, these entries allow users to identify the lowest flood magnitude preserved at a site, as well as the duration of the threshold, which corresponds to the time span between the oldest and youngest flood deposits at that location.

## 4 Comparison with other Databases

Table 7 compares seven palaeoflood records datasets published to date and one historical flood records database (Archer et al., 2019). This comparison highlights key differences in accessibility and the variety of record types. Although all databases provide temporal information and geolocated records, only PaleoRiada and Archer et al., (2019) stand out by providing easily accessible, publicly available data in a user-friendly open format. Moreover, both databases have open access to event magnitude data, a feature absent in the other databases. The implementation method of PaleoRiada is comparable in terms of completeness and accessibility to the historical flood records database published by Archer et al. (2019), demonstrating its applicability in multi-record flood analyses.

Regarding Spain, PaleoRiada offers broader spatial coverage, a wider variety of record types, and greater accessibility than earlier databases such as SPHERE-GIS and PaleoTagus. Concerning recently published global or international palaeoflood databases (Wilhelm et al., 2019 and 2022), PaleoRiada follows a similar conceptual model and includes equivalent fields and categories (https://wiki.linked.earth/File:FloodParameters.png). Like the PAGES-Floods Working Group Database, PaleoRiada allows for collaborative updates, facilitating the inclusion of new data. Furthermore, it adheres to the FAIR Guiding Principles—ensuring findability, accessibility, interoperability, and reusability (Wilkinson et al., 2016). All these factors contribute to PaleoRiada's dataset being potentially incorporable into global databases and usable by a broader range of user groups, including planning administrations and flood risk managers.

Table 7. Comparison of the general characteristics of palaeoflood records datasets. X: not available; G: global; N: national; R: regional.

| Characteristics                                 | Enzel<br>(1993) | Hirschboeck<br>et al., (1996) | Kale<br>(2008) | Wilhem<br>et al.,<br>(2019) | Greenbaum<br>et al.,<br>(2000) | SPHERE-<br>GIS and<br>PaleoTagus | PaleoRiada   | Archer et al., (2019)* |
|-------------------------------------------------|-----------------|-------------------------------|----------------|-----------------------------|--------------------------------|----------------------------------|--------------|------------------------|
| Spatial coverage                                | (R)<br>U.S.A    | (N)<br>U.S.A                  | (N)<br>India   | (G)                         | (R)<br>Israel                  | (R)<br>Spain                     | (N)<br>Spain | (N)<br>England         |
| Multiple palaeoflood record types               | X               | ✓                             | ✓              | ✓                           | ✓                              | X                                | ✓            | -                      |
| Temporal data                                   | ✓               | ✓                             | ✓              | ✓                           | $\checkmark$                   | $\checkmark$                     | $\checkmark$ | $\checkmark$           |
| Dating methods information                      | X               | ✓                             | ✓              | ✓                           | ✓                              | ✓                                | ✓            | <b>√</b>               |
| Geolocated records                              | $\checkmark$    | ✓                             | ✓              | ✓                           | $\checkmark$                   | $\checkmark$                     | ✓            | ✓                      |
| Easy to consult data (Findable Data)            | X               | X                             | X              | X                           | X                              | X                                | ✓            | ✓                      |
| Publicly accessible age data (per record)       | X               | X                             | X              | ✓                           | X                              | X                                | <b>√</b>     | <b>√</b>               |
| Publicly accessible magnitude data (per record) | X               | X                             | X              | X                           | X                              | X                                | ✓            | ✓                      |
| Accessible metadata                             | X               | ✓                             | X              | ✓                           | X                              | X                                | ✓            | $\checkmark$           |
| Open data format<br>(Interoperable Data)        | X               | X                             | X              | ✓                           | X                              | X                                | ✓            | ✓                      |
| Specified data source (Reusable Data)           | ✓               | X                             | ✓              | ✓                           | <b>√</b>                       | <b>√</b>                         | <b>√</b>     | <b>√</b>               |
| Collaborative Updating                          | X               | X                             | X              | ✓                           | X                              | X                                | ✓            | ✓                      |

<sup>\*</sup>Database of historical flood records. Public accessibility refers to the ease of download and usability of the dataset.

## 5 PaleoRiada database utility in the Spanish Flood-prone Mapping Project (SNCZI)

The European Flood Directive 2007/60/EC required the development of country-specific flood risk maps. In Spain, these maps were created through the Spanish Flood-prone Mapping Project (SNCZI, Sistema Nacional de Cartografía de Zonas Inundables (https://www.miteco.gob.es/es/agua/temas/gestion-de-los-riesgos-de-inundacion.html)), which uses flow data (derived from the CauMAX software, Jiménez Álvarez et al., 2013) but does not incorporate palaeoflood information. To assess the potential utility of PaleoRiada data for enhancing SNCZI flood maps, spatial analyses were conducted, intersecting the georeferenced PaleoRiada data with the flood zones mapped in SNCZI for return periods of 10, 100, and 500 years.

Results showed an 11-20% overlap of PaleoRiada sampling sites with SNCZI flood zones, increasing to 20-28% when a 100-metre buffer was applied. Most of the PaleoRiada data comes from small drainage basins, with 56 of 80 sampling sites having upstream areas of less than 1,250 km². This suggests that the PaleoRiada dataset could help locate new potentially flood-prone areas, serving as a basis for proposing preliminary flood risk studies (specifically in small ungauged basins). On the other

hand, in approximately 30% of PaleoRiada sites, major river confluences fall within the drainage distance between the palaeoflood record locations and urban areas, limiting the direct application of discharge data for flood hazard analysis in nearby urban centres.

A comparison of estimated discharges from PaleoRiada and CauMAX revealed that 14 sampling sites exceeded the 500-year CauMAX return period, 6 fell between the 100- and 500-year return periods, and 14 were below the 100-year return period. In this context, PaleoRiada data remains valuable for enhancing flood risk assessments, particularly in improving peak flow estimates.

## **6 Applications**

#### 6.1. Summary discussion

The main contribution of PaleoRiada is extending the temporal range of flood records in Spain, complementing instrumented records (Menéndez Campo and Quintas Ripoll, 1991) and historical datasets (Gil-Guirado et al., 2019; Llasat et al., 2009). While some debate the inclusion of palaeoflood data in risk analyses (Ballesteros-Cánovas et al., 2015b), it has been shown to have a significant impact on flood peak-flow and return period estimates (Ballesteros-Canovas et al., 2011a, 2013, 2016; Benito et al., 2021, 2023b) and flood hazard mapping (Garrote et al., 2018). Longer flood records are advantageous (Macdonald and Black, 2010; Greenbaum et al., 2014) and can maximise the contribution of palaeoflood data to flood frequency analysis, thereby significantly enhancing its reliability (Lam et al., 2017; Zituni et al., 2021; Benito et al., 2022).

PaleoRiada data are useful for ungauged basins, where they can aid in calibrating hydrological models (Sivapalan, 2003; Hrachowitz et al., 2013; Efstratiadis et al., 2014). They are also valuable for basins with relatively short instrumental records, ranging from a few decades to around a century (Greenbaum et al., 2014). The PaleoRiada dataset could also help analyse climate-flood relationships across warming and cooling periods (Wilhelm et al., 2022; Díez-Herrero et al., 2024; Benito et al., 2008; Moreno et al., 2008) or be evaluated with climate models for extreme flood projections (Baker et al., 2022), despite uncertainties over long-term climate variations (Bothe and Zorita, 2021; Amrhein et al., 2020). PaleoRiada has potential in international policy contexts like the Sendai Framework and the EU Directive 2007/60/EC, particularly for integrating climate change into flood risk assessments (Ely et al., 1991; Benito et al., 2005) and for infrastructure planning (Baker et al., 2022; Díez-Herrero, 2021). It may also assist in protecting historical and natural heritage (Garrote et al., 2022) and geoarchaeological studies (Jean-François, 2011; Wu et al., 2017). Beyond these applications, the PaleoRiada dataset could also be valuable for estimating transmission losses along channels, a key hydrological parameter that links surface flows to groundwater recharge, representing an essential component of water resources, particularly under changing climatic conditions (Greenbaum et al., 2002; Benito et al., 2011).

The future application of PaleoRiada will rely on continuous improvement and updating of the dataset. Key objectives will include identifying new palaeoflood sites, assessing their potential to provide valuable palaeohydrological data, and enhancing data entry methods and user participation through the web application.

## 6.2. Examples of application of PaleoRiada in Updating Preliminary Flood Risk Assessments

The PaleoRiada database is a valuable resource for updating Preliminary Flood Risk Assessments (www.miteco.gob.es/EPRI) under the European Directive 2007/60/EC, particularly for identifying or redefining Potential Significant Flood Risk Areas (PSFRA) (www.miteco.gob.es/ARPSI). In the following sections, we suggest three applications of PaleoRiada in Spain (Fig. 360 6).

#### 6.2.1 Modification of PSFRA Boundary (Guadalquivir River Basin)

In the Guadalquivir River Basin, the PSFRA in the Guadix Town (Granada) was initially defined based on the 1973 flood event of the Guadix River. However, PaleoRiada reveals two palaeoflood events (Roman period, 1st century AD, and Almohad period, 12th century AD) located just outside the current PSFRA boundary. These events, caused by the Almorejo stream (Díez-Herrero et al., 2024), could be considered to adjust the PSFRA boundary (including this tributary), reflecting a more accurate flood risk based on long-term flood patterns (**Fig. 6-A**).

## 6.2.2 Extension of PSFRA (Duero River Basin)

In the Duero River Basin, the PSFRA along the Eresma River (Segovia) is currently based on historical flooding near significant buildings. PaleoRiada, however, documents Pleistocene palaeofloods downstream, with discharges four times higher than recorded maxima. This suggests extending the PSFRA downstream to incorporate these ancient floods, highlighting significant hydrological changes during interglacial periods (Arsuaga et al., 2012) that could influence future flood risks (Fig. 6-B).

#### 6.2.3 Creation of a new PSFRA (Tajo River Arroyo Cabrera Basin)

In the Tajo River Basin, the Arroyo Cabrera stream (an Alberche River tributary) shows multiple palaeoflood records in a vulnerable area with historical impacts on local infrastructure (Ballesteros-Cánovas et al., 2011b). Based on these findings from PaleoRiada, a new PSFRA could be defined to improve the scope of Preliminary Flood Risk Assessments in small mountain basins (**Fig. 6-C**).

In practice, PaleoRiada has already been utilised by basin authorities responsible for reviewing and updating the Preliminary Flood Risk Assessments, aiding in the revision of PSFRA in various river basins across Spain. Notably, in the Tajo and Guadalquivir basins, authorities have systematically analysed the database records to determine whether to extend, adjust, or create a new PSFRA (https://www.chguadalquivir.es/EPRI).

Figure 6. Examples of PaleoRiada's application in identifying or redefining Potential Significant Flood Risk Areas (PSFRA) in Spain. A. Modification of PSFRA Boundary in the Guadix River, B. Extension of PSFRA in the Eresma River, C. Creation of a new PSFRA in the Arroyo Cabrera stream. Flood zones from https://sig.miteco.gob.es/snczi/ and base map from PNOA.

## 7 Conclusions

PaleoRiada is the first national database of published palaeoflood records with local applicability. Its public accessibility, ease of use, and reusability offer valuable opportunities for incorporating palaeoflood data into land-use planning and flood risk management, including identifying new areas for preliminary flood risk assessment and enhancing flood frequency analysis. The PaleoRiada dataset has 299 palaeoflood records (157 with discharge values), spans 97,000 years BP to 2014 CE, and includes diverse evidence types (geological and biological) across most Spanish river basins. PaleoRiada has spatial, temporal, typological, and magnitude biases due to research project distribution, evidence preservation and varied discharge estimation methods. Analysis of the database shows that fluvial sedimentological evidence provides the longest temporal perspective and preserves some of the oldest palaeoflood events. Although these records are associated with higher age uncertainties, they remain essential for understanding long-term flood variability.

The discharge values range from 0.02 to 320 m³·km²·s¹¹ and must be interpreted with respect to basin size, while also considering flow and flood types, such as hyperconcentrated flows, dam failures, or flash floods. While PaleoRiada contains extensive palaeoflood data, further analysis is needed to correlate records and identify individual flood events. Reporting of age and discharge uncertainties allows users to filter records by precision, ensuring they can be reliably integrated into hydrological studies and temporal analyses of flood frequency and occurrence. Minimum discharge values, together with contextual and stratigraphic information, allow users to identify valuable perception thresholds for flood frequency analyses. Despite the availability of palaeoflood databases, their use in flood risk analysis remains limited due to low data accessibility, the absence of peak flow values, and the reliance on conventional data sources. Increased efforts are needed to collect palaeoflood data, especially in high-risk areas, and to standardise its use in flood risk assessments.

Data availability. The PaleoRiada Database, including the data and metadata, are available for download in Zenodo at https://zenodo.org/records/17391823 (Sandoval-Rincón et al., 2025).

- Author contributions. K. P. Sandoval-Rincón: investigation, formal analysis and writing original draft. J. Garrote-Revilla: investigation and writing original draft. S. Cervel and J. Hernández-Manchado: methodology. J. López-Vinielles; D. Vázquez-Tarrío, J. A. Ballesteros-Cánovas, G. Benito and R. M. Mateos: investigation and writing review and editing. A. Díez-Herrero: conceptualisation, funding acquisition, investigation, writing, review and editing. Competing interests: The authors declare that they have no conflict of interest.
- Acknowledgements. The authors would like to thank the collaboration provided by various people, grants, and institutions, including, the following: Predoctoral research grant PIPF-2022/ECO-24879, funded by Comunidad de Madrid. Two grants (2022-2023 and 2023-2026) which include actions 20223TE003 and 20233TE012 (Tarquín project, IGME-CSIC) signed between Dirección General del Agua (DGA-MITERD) and Consejo Superior de Investigaciones Científicas (CSIC). Tarquín project colleagues: Mario Hernández, Ana Lucía Vela and M. Ángeles Perucha. The authors especially thank Jhonatan Rivera for his collaboration in designing the PaleoRiada web application.

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
