# Peer review of "PaleoRiada: A new integrated spatial database of palaeofloods in Spain"

_Earth System Science Data, 2024_

## Referee Comment (RC1)

In addition to being a valuable dataset for hydrologists and hazard managers in Spain, PaleoRiada represents a significant improvement in the design of paleoflood databases. This database addresses the shortfalls of many of the existing global databases that limit their applicability to the wider hydrological and hazard mitigation communities. To my knowledge, no other publicly available database contains a comparable level of detail concerning types of floods or quantitative hydrological data (stage, discharge, etc.). I can attest to the ease of access of the data through both Zenodo and ArcGIS online web application, having tested both.

**Broader conceptual and methodological questions.**

1.  Table 3 shows the distribution of records based on the type of evidence used to reconstruct paleofloods:

Can you provide more detail in the table and in the text about what constitutes the "erosive" forms evidence sub-sub-type? I can think of some possibilities (stripped floodplains; truncated soils and sediments, for example), but I am unclear whether these apply.

2.  Of the types of paleoflood evidence used, there is no mention in tables or text or the database of non-exceedance data, which are often collected in paleoflood studies to help with determining upper hydrologic bounds and perception thresholds. Were these data included?

3.  In a related question, in cases where stage and/or discharge data were determined (n=197) were perception thresholds used and are they included in the database, such as in "other hydrological interpretation data?" I ask because these data would be beneficial for flood frequency analyses that may incorporate data from the database.

4.  Regarding age controls applied to the paleofloods, was there any restriction on including paleofloods with age estimates with too large an error range? I ask because in researching published paleoflood records many contain very large error ranges. Their inclusion in temporal analyses of flood frequency or occurrence my artificially increase the total number. I think the paper would benefit from some discussion of the range of age estimate uncertainties of the paleofloods included in the database. Are the error estimates greater in a particular basin or type of paleoflood evidence, for example? This could help a user help make decisions about the applicability of the data for their purposes.

5.  A related question is how radiocarbon calibration curve differences are handled? Over time, the calibration curve has changed (Intecal 19 vs Intecal 20). Given that a large percentage of the database is comprised of paleofloods dated in the Common

Era with radiometric dating (38%, Table 2), it should be explained and acknowledges in the paper that differences in radiocarbon ages can be caused by the application of different calibration curves. I also suggest that the calibration curve used in cases of radiocarbon-dated paleofloods be retrievable in the database. This would allow someone planning to use the data to improve older radiocarbon age-estimates with a new calibration curve.

**Technical comments**

1. Line 26: Please clarify if these are all overbank floods or do they also include pluvial flood events

2. Line 208: "Erosive forms" type

3. Table 3: "G" in geomorphological needs capitalizing to be consistent with other types in the column

4. Figure 4 caption: "records" needs an apostrophe to make it plural and possessive "records'"

5. Fig. 5: center the axis labels

6. Instrumental is often used in the paper to refer to the gauged record of streamflow but this word, strictly speaking is an adjective meaning "essential to an outcome" or "something musical." Suggest replacing the word "instrumental" with "instrumented or instrumentation." Instrumented record is better suited to describing gauged flow or something measured with a scientific instrument.

7. Lines 35, 39

8. Line 53 Spelling of flood frequency: remove dash

9. Sentences that should have a colon before numbered list:

10. Lines 39, 59, 99; 220

11. Lines 148-149 has spacing issues because of the paragraphs have justified alignment; switch this line to left alignment, if needed.

12. Line 159: insert comma before "such as"

13. Fig. 3 needs a label for the "x" axis

14. Line 195: Rewrite sentence to not begin with the word "or."

15. Line 196-198 is one very long sentence that is hard to read. I suggest breaking this single sentence up into several shorter ones or use fewer words in the phrasing

16. Line 202: no need for comma after "sites"

17. Line 204: delete "s" in "plains" to make it singular

18. Line 228: extra space in"torrents"

19. Line 231: change phrasing to "the spatiotemporal distribution of discharge"

20. Table 5, unnecessary comma after "dataset"

21. Line 250 unnecessary comma before parenthesis

22. Line 310: re-write last sentence to not have a colon, as there are sections that follow and not items in a series. "In the following sections, we suggest three applications of PaleoRiada in Spain."

23. Line 343: add s to "values"

---

## Community Comment (CC1)

We thank the reviewer for her thorough assessment of the manuscript and for the constructive suggestions. These comments have helped us to improve the clarity, consistency, and completeness of the work. Below, we provide detailed responses to each point and indicate the corresponding changes made in the manuscript.

**Q1.** We have replaced the sub-sub-type name "erosive forms" with "erosional scars" to be more specific. In the revised text and Table 4, this category now explicitly refers to geomorphological features resulting from flood-induced erosion, such as sigmoid longitudinal profiles and bank undercuttings. We have also included references (e.g., Bodoque et al., 2011; Jarrett and Tomlinson, 2000) so that readers can consult these works for a more detailed explanation of each type of evidence mentioned.

**Q2.** Regarding non-exceedance bounds (time intervals during which palaeostages have not exceeded the level needed to modify a terrace or high surface; Levish, 2002), we agree that it would be valuable to add a field for non-exceedance discharges (NEB). However, only a few studies of Spanish palaeoflood records mention possible non-exceedance evidence, such as the absence of depositional units or the preservation of old soils (indicating no erosion) on high terraces or surfaces.

With respect to perception thresholds, these were not separately coded because their value depends strongly on the depositional setting. In some contexts, the threshold can remain essentially fixed (e.g., sediment accumulation inside small caves), while in others it rises as successive deposits build up on flood benches. Nevertheless, perception thresholds can be derived from the database by combining the reported "minimum discharge" values (RQI field in the Hydrological Information Table) with the "other information" field (OE in the Record Basic Data Table), which describes the stratigraphic setting. In practice, the minimum discharge represented by a set of flood beds at a given site defines the perception threshold, and the duration of that threshold corresponds to the time span between the oldest and youngest flood ages at that site.

Levish, D.R., 2002. Paleohydrologic Bounds: Nonexceedance information for flood hazard assessment. In: House, P.K., Webb, R.H., Baker, V.R., Levish, D.R. (Eds.), *Ancient Floods, Modern Hazards: Principles and Applications of Paleoflood Hydrology*, Water Science and Application Series, vol. 5. American Geophysical Union, Washington, DC, pp. 175–190.

**Q3.** In the reviewed literature, explicit reporting of perception thresholds is generally limited to a few studies focused on flood frequency analysis. In contrast, studies primarily addressing flood–climate variability rarely include them. As noted in our previous reply to Q2, however, perception thresholds can still be inferred from several fields in the database. In practice, they can be extracted by combining the "minimum discharge" values with contextual information in "other hydrological interpretation data" and stratigraphic descriptions. Thus, while not explicitly coded

as a separate field, the database contains the necessary information to reconstruct perception thresholds for use in flood-frequency analyses.

In the updated version of the manuscript, at the end of Section 3.3 Hydrological data, the following text has been added:

"Information on non-exceedance bounds and perception thresholds can be derived from the database even though the latter is not coded as a separate field. Non-exceedance bounds are only occasionally reported in the reviewed studies, usually inferred from the absence of depositional units or the preservation of stable soils on high terraces or surfaces. Although not systematically available, such evidence highlights intervals when flood stages did not exceed certain geomorphic thresholds, and it is documented in the Hydrological Information Table via the NEB field.

Perception thresholds, by contrast, can be reconstructed for most sites. These thresholds represent the minimum discharge or stage required to leave identifiable flood evidence at a depositional setting, and in some cases, they increase through time as sediment progressively builds up on flood benches (a self-rising component). In practice, perception thresholds can be obtained by combining the "minimum discharge" field with the contextual information provided in "other hydrological interpretation data" and the stratigraphic descriptions. Together, these entries allow users to identify the lowest flood magnitude preserved at a site, as well as the duration of the threshold, which corresponds to the time span between the oldest and youngest flood deposits at that location."

**Q4**. In the updated version of the manuscript (Section 3.1, Temporal data), we have included a table summarising the general statistics of age-uncertainty values in the PaleoRiada dataset, highlighting their relationship with both, the age of the palaeoflood evidence and the dating methods used. The discussion emphasises that age uncertainty is strongly influenced by the antiquity of the records, with the highest values associated with fluvial sedimentological evidence—primarily because these records include some of the oldest events preserved in durable geological deposits. It is also clarified that no records have been excluded due to high uncertainty, to maintain the completeness of the dataset. Instead, users are encouraged to filter records according to the temporal precision required for their specific analyses.

Regarding the question, fluvial sedimentological records show the highest average age uncertainty (± 293.8 years), largely due to the antiquity of the oldest preserved events. Many were dated using luminescence, with uncertainties exceeding ± 3,000 years, reflecting the difficulty of constraining long-preserved deposits. Age uncertainty does not follow a consistent pattern across hydrographic basins; instead, it is shaped by past research locations, site accessibility, and preservation

conditions. Thus, uncertainty is better explained by evidence type and age than by basin.

**Q5.** Our review of the sources shows that, out of more than 300 records compiled, only approximately 20% explicitly report the calibration curve applied. In the remaining cases, the information was either not applicable to the dating method employed or was not reported in the original references.

To account for this, we have incorporated the available information on calibration curves into the OT field of the Record Basic Data table. This enables users to identify those cases where the calibration framework is known and, if desired, to recalibrate radiocarbon ages with updated curves such as IntCal20.

---

## Community Comment (CC2)

**Q1**.

We thank the reviewer for this comment. Information on river/channel forms has been included in the database within the Hydrological Information Table (HIT), specifically under the field STT (Stream Type), which makes it possible to identify palaeoflood evidence in different geomorphological settings such as torrential mountain streams, wide floodplain rivers, confined streams, and ephemeral Mediterranean streams. We have not incorporated this information into the Basic Information Tables (BIT), since these are intended to present only the fundamental descriptive data of each record.

**Q2.**

We agree with the reviewer that the type of evidence is a key element of palaeoflood studies. This information has been incorporated in the database within the *Record Basic Data table* (fields T, ST and SST). In addition, the article text provides a synthesis of this information (Table 4).

Q3.

We greatly appreciate this suggestion and fully agree with the reviewer. Frequency analyses based on the PaleoRiada database will inevitably reflect the same magnitude bias as the original studies from which the data were derived. This limitation is already acknowledged in the manuscript (Section 3.3 Hydrological Data).

---

## Author Comment (AC1)

We would like to thank you for carefully reading our manuscript and for your constructive comments and suggestions. We have revised the manuscript accordingly and believe that these changes have substantially improved the clarity of the database's applications and limitations. Below, we provide our responses to each of your comments.

**Major comments:**

**MC1:** We appreciate the reviewer's observation. To address this, we have added a sentence in the Hydrological data section highlighting the need to consider differences between past and present land use, as these may cause substantial variations in flood magnitudes within the same basin.

**MC2**: To avoid confusion, we have replaced the abbreviation $Q/A_{max}$ with $Qs_{max}$ (maximum specific discharge recorded at each sampling site). In addition, we have added a new column, "A", indicating the drainage area ($km^2$), and specified in the footnotes that this refers to the drainage surface area up to the sampling site.

**MC3 and MC4:** We have revised the manuscript to clarify that the PaleoRiada dataset is biased towards larger events, that discharge estimates may be under- or overestimated, and that long-term channel changes may affect the estimation of hydrological data. We also note that smaller and more frequent floods are typically absent, and these limitations should therefore be considered in flood frequency analysis.

**MC5:** We thank the reviewer for highlighting the importance of transmission losses and their role in linking surface flows to groundwater recharge. We have revised Section 6 ("Applications") to include this aspect.

All minor comments and corrections have been addressed in the revised manuscript. Specifically, the terminology regarding flood records has been clarified, and Table 6 has been updated to include the suggested records and references. Figure 6 captions have been amended to specify the relevant rivers, and references to Fig. 6a, 6b and 6c have been added to Sections 6.2.1–6.2.3 accordingly.

We are grateful for your valuable feedback, which has helped us strengthen the manuscript.

---

## Author Response (AR1)

**RESPONSE**

**(essd-2024-549)**

We thank the referees for their thorough assessment of the manuscript and for their constructive suggestions. These comments have helped us to improve the clarity, consistency, and completeness of the work. Below, we provide detailed responses to each point and indicate the corresponding changes made in the manuscript.

**Referee comment 1 (RC1)**

**01.**

Table 3 shows the distribution of records based on the type of evidence used to reconstruct paleofloods: Can you provide more detail in the table and in the text about what constitutes the "erosive" forms evidence sub-sub-type? I can think of some possibilities (stripped floodplains; truncated soils and sediments, for example), but I am unclear whether these apply.

**R1.**

We have replaced the sub-sub-type name "erosive forms" with "erosional scars" to be more specific. In the revised text and Table 4, this category now explicitly refers to geomorphological features resulting from flood-induced erosion, such as sigmoid longitudinal profiles and bank undercuttings. We have also included references (e.g., Bodoque et al., 2011; Jarrett and Tomlinson, 2000) so that readers can consult these works for a more detailed explanation of each type of evidence mentioned.

**Q2.**

Of the types of paleoflood evidence used, there is no mention in tables or text or the database of non-exceedance data, which are often collected in paleoflood studies to help with determining upper hydrologic bounds and perception thresholds. Were these data included?

**R2.**

Regarding non-exceedance bounds (time intervals during which palaeostages have not exceeded the level needed to modify a terrace or high surface; Levish, 2002), we agree that it would be valuable to add a field for non-exceedance discharges (NEB). However, only a few studies mention possible non-exceedance evidence, such as the absence of depositional units or the preservation of old soils (indicating no erosion) on high terraces or surfaces.

With respect to perception thresholds, these were not separately coded because their value depends strongly on the depositional setting. In some contexts, the threshold can remain essentially fixed (e.g., sediment accumulation inside small caves), while in others it rises as successive deposits build up on flood benches. Nevertheless, perception thresholds can be derived from the database by combining the reported "minimum discharge" values (RQI field in the Hydrological Information Table) with the "other information" field (OE in the Record Basic Data Table), which describes the stratigraphic setting. In practice, the minimum discharge represented by a set of flood beds at a given site defines the perception threshold, and the duration of that threshold corresponds to the time span between the oldest and youngest flood ages at that site.

Please note the text added on lines 80-84 of the revised manuscript.

**Ref.** Levish, D.R., 2002. Paleohydrologic Bounds: Nonexceedance information for flood hazard assessment. In: House, P.K., Webb, R.H., Baker, V.R., Levish, D.R. (Eds.), Ancient Floods, Modern Hazards: Principles and Applications of Paleoflood Hydrology, Water Science and Application Series, vol. 5. American Geophysical Union, Washington, DC, pp. 175–190.

**Q3.**

In a related question, in cases where stage and/or discharge data were determined (n=197) were perception thresholds used and are they included in the database, such as in "other hydrological interpretation data?" I ask because these data would be beneficial for flood frequency analyses that may incorporate data from the database.

**R3.**

In the reviewed literature, explicit reporting of perception thresholds is generally limited to a few studies focused on flood frequency analysis. In contrast, studies primarily addressing flood—climate variability rarely include them. As noted in our previous reply, however, perception thresholds can still be inferred from several fields in the database. In practice, they can be extracted by combining the "minimum discharge" values with contextual information in "other hydrological interpretation data" and stratigraphic descriptions. Thus, while not explicitly coded as a separate field, the database contains the necessary information to reconstruct perception thresholds for use in flood-frequency analyses.

In the updated version of the manuscript, at the end of Section 3.3 Hydrological data (lines 80-91), the following text has been added:

"Information on non-exceedance bounds and perception thresholds can be derived from the database even though the latter is not coded as a separate field. Non-exceedance bounds are only occasionally reported in the reviewed studies, usually inferred from the absence of depositional units or the preservation of stable soils on high terraces or surfaces. Although not systematically available, such evidence highlights intervals when flood stages did not exceed certain geomorphic thresholds, and it is documented in the Hydrological Information Table via the NEB field.

Perception thresholds, by contrast, can be reconstructed for most sites. These thresholds represent the minimum discharge or stage required to leave identifiable flood evidence at a depositional setting, and in some cases, they increase through time as sediment progressively builds up on flood benches (a self-rising component). In practice, perception thresholds can be obtained by combining the "minimum discharge" field with the contextual information provided in "other hydrological interpretation data" and the stratigraphic descriptions. Together, these entries allow users to identify the lowest flood magnitude preserved at a site, as well as the duration of the threshold, which corresponds to the time span between the oldest and youngest flood deposits at that location."

**Q4**.**

Regarding age controls applied to the paleofloods, was there any restriction on including paleofloods with age estimates with too large an error range? I ask because in researching published paleoflood records many contain very large error ranges. Their inclusion in temporal analyses of flood frequency or occurrence my artificially increase the total number. I think the paper would benefit from some discussion of the range of age estimate uncertainties of the paleofloods included in the database. Are the error estimates greater in a particular basin or type

of paleoflood evidence, for example? This could help a user help make decisions about the applicability of the data for their purposes.

**R4.**

In the updated version of the manuscript (Section 3.1: Temporal data, lines 197-217), we have included a table summarising the general statistics of age-uncertainty values in the PaleoRiada dataset, highlighting their relationship with both the age of the palaeoflood evidence and the dating methods used. The discussion emphasises that age uncertainty is strongly influenced by the antiquity of the records, with the highest values associated with fluvial sedimentological evidence—primarily because these records include some of the oldest events preserved in durable geological deposits. It is also clarified that no records have been excluded due to high uncertainty, to maintain the completeness of the dataset. Instead, users are encouraged to filter records according to the temporal precision required for their specific analyses.

Regarding the question, fluvial sedimentological records show the highest average age uncertainty ( $\pm$  293.8 years), largely due to the antiquity of the oldest preserved events. Many were dated using luminescence, with uncertainties exceeding  $\pm$  3,000 years, reflecting the difficulty of constraining long-preserved deposits. Age uncertainty does not follow a consistent pattern across hydrographic basins; instead, it is shaped by past research locations, site accessibility, and preservation conditions. Thus, uncertainty is better explained by evidence type and age than by basin.

The added text is specifically the following:

"The PaleoRiada dataset contains explicit numerical age-uncertainty values (in years) only for records dated using radiometric methods, luminescence techniques and lichenometry (**Table 3**). These uncertainties range from as little as ±4 years for very recent lichenometric or 210Pb-based dates (e.g. late 20th–21st century CE) to as much as ±6,000 years for the oldest OSL-dated records (44,736–97,000 BP). Radiocarbon (14C) dating, which in the database has been applied to both BCE and CE records, shows a comparable pattern: uncertainties are smallest (±30–95 years) for more recent samples (30–1,947 CE) and largest (±1,860 years) for older samples (2,640–32,600 BP). This indicates that, within the dataset, age uncertainty is governed primarily by the antiquity of the sample rather than by the dating method alone. Users requiring high temporal precision can make use of the uncertainty field (IDT in the Record Basic Data Table) to filter and select the most appropriate records for frequency or occurrence analyses. Notably, no records were excluded from PaleoRiada due to high age uncertainty, to preserve the completeness of the published evidence; instead, uncertainty information is provided so that each user may apply selection criteria suited to their specific needs".

**Q5.**

A related question is how radiocarbon calibration curve differences are handled? Over time, the calibration curve has changed (Intecal 19 vs Intecal 20). Given that a large percentage of the database is comprised of paleofloods dated in the Common Era with radiometric dating (38%, Table 2), it should be explained and acknowledges in the paper that differences in radiocarbon ages can be caused by the application of different calibration curves. I also suggest that the calibration curve used in cases of radiocarbon-dated paleofloods be retrievable in the database. This would allow someone planning to use the data to improve older radiocarbon age-estimates with a new calibration curve.

**R5.**

Our review of the sources shows that, out of more than 300 records compiled, approximately 20% explicitly report the calibration curve applied. In the remaining cases, the information was either not applicable to the dating method employed or was not reported in the original references.

To account for this, we have incorporated the available information on calibration curves into the OT field of the Record Basic Data table. This enables users to identify those cases where the calibration framework is known and, if desired, to recalibrate radiocarbon ages with updated curves such as IntCal20.

**Referee comment 2 (RC2)**

PaleoRiada database, through the conceptual, logical, and physical models, systematically integrated the paleoflood records of Spain and made accessible to users through forms such as web GIS. It also served the local water management. I believe such a database is not only necessary for analyzing the relationship between climate and flood disasters, but also very important for modern flood risk management of water systems.

I have some minor questions or suggestions:

**Q1.**

In BIT data, have you included the river/channel forms of paleoflood? For examples, did these paleoflood occur in narrow valley, wide channel or river beach?

**R1.**

We thank the referee for this observation. Information on river/channel forms has been included in the database within the Hydrological Information Table (HIT), specifically under the field STT (Stream Type), which makes it possible to identify palaeoflood evidence in different geomorphological settings such as torrential mountain streams, wide floodplain rivers, confined streams, and ephemeral Mediterranean streams. We have not incorporated this information into the Basic Information Tables (BIT), since these are intended to present only the fundamental descriptive data of each record.

**Q2**.**

The evidence of each paleoflood (sediments, channel erosion, vegetation damage, etc) should be mentioned in tables or text or the database, to help the later researchers.

**R2.**

We agree with the referee that the type of evidence is a key element of palaeoflood studies. This information has been incorporated in the database within the *Record Basic Data table* (fields T, ST and SST). In addition, the article text provides a synthesis of this information (**Table 4**).

**Q3.**

I suggest that you mention the limitation somewhere, as some evidence of early paleoflood probably were destroyed by later more strong flood, this will influence the frequency analyses.

**R3**.**

Frequency analyses based on the PaleoRiada database will inevitably reflect the same magnitude bias as the original studies from which the data were derived. This limitation is already acknowledged in the revised manuscript (Section 3.3: Hydrological Data, lines 264-270).

The text is specifically the following:

"It is worth noting that the PaleoRiada dataset has a flood event magnitude bias. Over- or underestimation of discharge values may occur for several reasons. Many discharge estimates correspond to minimum water levels that could have been reached (e.g. slackwater deposits and tree scar records). Palaeoflood evidence is preferentially preserved for the largest extreme floods (Thorndycraft and Benito, 2006), meaning that smaller and more frequent events are typically absent. For long records, channel bed or base levels may have changed; however, most records in PaleoRiada are derived from relatively stable river sections, minimising this effect. These limitations should therefore be considered when using the PaleoRiada dataset for hydrological analyses"

**Referee comment 3 (RC3)**

**Q1.**

Lines 183-185 – Similar to the comment about long records and past climate, the issue of past land-use versus present land-use, which may cause large differences for the same basin, should be mentioned. A sentence/section related to this, is clearly needed.

**R1.**

We appreciate the referee's observation. To address this, we have added a sentence in the 3.3 Hydrological data section (lines 270-272) highlighting the need to consider differences between past and present land use, as these may cause substantial variations in flood magnitudes within the same basin.

The added text is specifically the following:

"Similarly, when using the PaleoRiada discharge data for long-term analyses of floods and climate variability, it is essential to consider the differences between past and present land-use, which may cause substantial variations in flood magnitudes within the same basin (Feinberg et al., 2020; Schillereff et al., 2019)."

**Q2.**

Table 4-(a) What is the  $A_{max}$ ? Is it the area of the entire basin or the area up to the measuring/study site? (b) Although the table uses specific peak discharges Q/A the enormous range -3-4 orders of magnitude, needs some clarification. It is hard to compare a small arroyo to the Duero, for example. I suggest to add a column of the basin area to each basin.

**R2.**

To avoid confusion, we have replaced the abbreviation  $Q/A_{max}$  with  $Qs_{max}$  (maximum specific discharge recorded at each sampling site) in Table 5 (before Table 4). In addition, we have added

a new column, "A", indicating the drainage area (km²), and specified in the footnotes that this refers to the drainage surface area up to the sampling site.

**Q3.**

Lines 238-240 – Although paleoflood records in rivers are considered conservative because of the gap between the related sediments deposition and the water elevation, for long records levels of the channel bed or base levels may have changed. These may cause over/under estimation of discharges and is a limitation which causes uncertainty. I believe it has to be clarified

\* Paleoflood records are usually partial records depending on the preservation of the evidence. Usually these records include the largest floods in term of discharge but misses the small and more frequent floods. This has to be taken into consideration in any analysis for any purpose when using these data.

**R3.**

We have revised the manuscript to clarify that the PaleoRiada dataset is biased towards larger events, that discharge estimates may be under- or overestimated, and that long-term channel changes may affect the estimation of hydrological data. We also note that smaller and more frequent floods are typically absent, and these limitations should therefore be considered in flood frequency analysis.

The added text is specifically the following:

"It is worth noting that the PaleoRiada dataset has a flood event magnitude bias. Over- or underestimation of discharge values may occur for several reasons. Many discharge estimates correspond to minimum water levels that could have been reached (e.g. slackwater deposits and tree scar records). Palaeoflood evidence is preferentially preserved for the largest extreme floods (Thorndycraft and Benito, 2006), meaning that smaller and more frequent events are typically absent. For long records, channel bed or base levels may have changed; however, most records in PaleoRiada are derived from relatively stable river sections, minimising this effect. These limitations should therefore be considered when using the PaleoRiada dataset for hydrological analyses."

**Q4**.**

In addition to risk assessment and climate change a very important implication of paleofloods is estimations of transmission losses along channels. This important hydrological parameter connects surface flows and groundwater recharge which is a major component of water resources mainly during climate changes. For this issue see: Greenbaum et al. (2002); Dahan et al. (2008); Morin et al. (2009); Benito et al. (2010; 2011).

**R4.**

We thank the referee for highlighting the importance of transmission losses and their role in linking surface flows to groundwater recharge. We have revised Section 6: Applications (lines 349-352) to include this aspect.

The added text is specifically the following:

"Beyond these applications, the PaleoRiada dataset could also be valuable for estimating transmission losses along channels, a key hydrological parameter that links surface flows to

groundwater recharge, representing an essential component of water resources, particularly under changing climatic conditions (Greenbaum et al., 2002; Benito et al., 2011)."

All minor comments and corrections have been addressed in the revised manuscript. Specifically, the terminology regarding flood records has been clarified, and Table 6 has been updated to include the suggested records and references. Figure 6 captions have been amended to specify the relevant rivers, and references to Fig. 6a, 6b and 6c have been added to Sections 6.2.1–6.2.3 accordingly.

We are grateful for your valuable feedback, which has helped us strengthen the manuscript.